# Pallister–Killian Syndrome versus Trisomy 12p—A Clinical Study of 5 New Cases and a Literature Review

**DOI:** 10.3390/genes12060811

**Published:** 2021-05-26

**Authors:** Aurora Arghir, Roxana Popescu, Irina Resmerita, Magdalena Budisteanu, Lacramioara Ionela Butnariu, Eusebiu Vlad Gorduza, Mihaela Gramescu, Monica Cristina Panzaru, Sorina Mihaela Papuc, Adriana Sireteanu, Andreea Tutulan-Cunita, Cristina Rusu

**Affiliations:** 1“Victor Babes” National Institute of Pathology, 050096 Bucharest, Romania; aura_arghir@yahoo.com (A.A.); ela_papuc@yahoo.com (S.M.P.); andreea_cunita@yahoo.com (A.T.-C.); 2Department of Medical Genetics “Saint Mary” Emergency Children’s Hospital, St. Vasile Lupu No 62, 700309 Iasi, Romania; lacrybutnariu@gmail.com (L.I.B.); monica.panzaru@umfiasi.ro (M.C.P.); abcrusu@gmail.com (C.R.); 3Department of Medical Genetics, Faculty of Medicine, “Grigore T. Popa” University of Medicine and Pharmacy, University Street, No 16, 700115 Iasi, Romania; vgord@mail.com (E.V.G.); mihaelagramescu@yahoo.ro (M.G.); 4Obregia Clinical Hospital of Psychiatry, 041914 Bucharest, Romania; magda_efrim@yahoo.com; 5Regional Institute of Oncology, 700483 Iasi, Romania; adryas@gmail.com; 6Cytogenomic Medical Laboratory, 14453 Bucharest, Romania

**Keywords:** Pallister Killian syndrome, trisomy 12p, mosaic, MLPA, array CGH

## Abstract

Pallister–Killian syndrome (PKS) is a rare, sporadic disorder defined by a characteristic dysmorphic face, pigmentary skin anomalies, intellectual disability, hypotonia, and seizures caused by 12p tetrasomy due to an extra isochromosome 12p. We present three cases of PKS and two cases of trisomy 12p to illustrate and discuss features rarely cited in the literature, present certain particularities that not yet been cited, and analyze the differences between entities. Moreover, we present alternative methods of diagnosis that could be easily used in daily practice. Features not yet or rarely reported in PKS literature include marked excess of hair on the forehead and ears in the first months of life, a particular eye disorder (abnormal iris color with pointed pupil), connective tissue defects, repeated episodes of infection and autonomic dysfunction, endocrine malfunction as a possible cause of postnatal growth deficit, more complex sensory impairments, and mild early myoclonic jerks. After performing different combinations of tests, we conclude that MLPA (follow-up kit P230-B1) or array CGH using DNA extracted from a buccal swab is a reliable method of diagnosis in PKS and we recommend either one as a first intention diagnostic test. In cases without major defects associated (suspicion trisomy 12p), subtelomeric MLPA should be performed first.

## 1. Introduction

Pallister–Killian syndrome (PKS, OMIM #601803) is a rare, sporadic genetic disorder defined by the association of a characteristic dysmorphic face with pigmentary skin anomalies, profound intellectual disability, hypotonia, and seizures [1,2,3]. The disorder was described first in adults by Pallister [4] and later in children by Killian and Tescler-Nicola [5], and shows a moderate preponderance of females [6]. It has a prevalence of 1/20,000 and is due to mosaic 12p tetrasomy resulting from a supernumerary isochromosome i(12p) [6,7,8] and only rarely due to mosaic 12p hexasomy with two isochromosomes i(12p) [7,9]. Some early cases were misinterpreted as mosaic tetrasomy 21q [10] or trisomy 12p with proximal 12q [6]. Tissue-specific mosaicism is characteristic for this condition—the detection rate of 12p tetrasomy is 0–2% in lymphocytes, 50–100% in fibroblasts and chorionic villi, and 100% in amniocytes and bone marrow cells [11]. There are a few possible explanations for the low detection rate of the mosaicism in lymphocytes; either the abnormal cells do not respond to the PHA stimulation used for cell culture [12], or there is an in vivo and in vitro selection against i12p positive cells [13]. To overcome these impediments, new diagnostic methods have been applied, including interphase FISH on lymphocytes, skin fibroblasts, or a buccal smear [13,14]; chromosomal microarray analysis [15,16]; or droplet digital PCR, which detects low levels of mosaicism [16]. The SNP array also evaluates meiotic origin [15], whereas MLPA is a simple, low-cost, multiplex diagnostic method with high accuracy in copy number estimation [17]. In most cases, the isochromosome is inherited from the mother (with the underlying mechanism thought to involve a combination of centromere maldivision and non-disjunction at meiosis) [15,18], and the occurrence of the disorder seems to be related with advanced maternal age [19]. Furthermore, i12p has been reported to be unstable with age in blood [16] but constant in skin fibroblasts [15,20]. The critical region seems to be 12p13.31, with the strongest candidate genes being *ING4*, *CHD4*, and *MAGP2* [16,20].

Typical facial characteristics include frontal bossing and high frontal hairline, temporofrontal balding, sparse eyebrows and lashes, hypertelorism, small and flat nose, full cheeks, long and simple philtrum, large mouth with downturned corners, thin upper lip and distinct cupid bow (“Pallister lip”), micrognathia, and low-set dysplastic ears [21,22,23,24,25,26,27,28,29,30]. Orodental features also include macroglossia, alveolar ridge or gingival overgrowth, delayed teeth eruption, and missing or double teeth [28]. As the child grows older, baldness diminishes (wiry hair or hair that grows only a few centimeters long covers the frontotemporal regions), the face becomes coarser, the eyebrows become thick, macroglossia develops, and the lips and chin become prominent [26,31,32]. Skin depigmentation is usually patchy, suggesting the presence of a mosaic, but is not always visible (Wood lamp examination may be indicated if suspicion of PKS is present). Some cases have limb shortening (rhizomelic or proportionate), lymphedema, and increased soft tissues of the extremities, but commonly palms and fingers are short and broad, with transverse flexion creases and clinodactyly 5 [25,29]. The most frequent malformations associated are supernumerary nipples, congenital heart defects, congenital diaphragmatic hernia, and anal defects. Renal, genital, and ocular malformations are rarely associated [21]. Marked hypotonia (causing feeding difficulties) and profound intellectual disability are present from birth onwards, with developing seizures and contractures. However, mild–moderate ID cases have been cited, as well as self-stimulatory and self-injurious (hand biting) behavior [29]. Seizures (myoclonic, generalized tonic–clonic, clustered tonic spasms, and absence) are common, have variable age of onset (typically occur in the first four years of life, not neonatal), and are frequently associated with non-epileptic paroxysmal events, which require combined therapies. Subjacent pathomechanisms include disturbances of ligand- or voltage-gated ion channels, synaptic function, or brain development [27,33]. Brain MRI reveals brain atrophy, corpus callosum dysgenesis, polymicrogyria, and spot calcifications in the perisylvian region [34]. Tethered cord has been found in some cases [35]. Sensory impairment (deafness and blindness) is frequently associated [35,36,37]. PKS probably includes autonomic dysfunction, anhidrosis or hypohidrosis, and episodes of hyperventilation interspersed with breath-holding being relatively common [29]. Growth has a very specific pattern—excessive prenatally (leading to macrosomia at birth) and postnatal decline in growth velocity. Anterior fontanel closure is delayed. Puberty is within the normal age ranges in girls and delayed in boys. The oldest patient reported was 45 years old [29]. The radiological examination may reveal ovoid vertebral bodies, delayed axial and pubic bone maturation, as well as a metaphyseal flare of long bones. Quantitative ultrasound is a radiation-free method that can be used to evaluate bone quality and abnormalities [28]. Differential diagnosis should consider mainly Fryns syndrome, Trisomy 12p, and Sifrim-Hitz-Weiss syndrome [3]. Next-generation phenotyping software using photoanalysis could be useful [32].

Almost all cases in the literature have been sporadic, with the recurrence risk being very low [34]. Fetal movements are decreased or late. PKS fetuses have a very typical growth pattern—increased biparietal diameter (BP) and head circumference (HC) above the 90th percentile associated with significant femoral growth delay under the 10th percentile [2]. Prenatal sonographic features suggestive for PKS include polyhydramnios, increased nuchal translucency, extremely flat facial profile, congenital diaphragmatic hernia, small stomach, as well as short limbs with small or abnormal extremities [11]. Ductus venosus agenesis might be a prenatal marker, however as it is associated with many complex syndromes, only its association with polyhydramnios, fetal macrosomia, and eventually short femoral bone or other minor defects should alert for PKS [30,38]. Array CGH on genomic DNA extraction from uncultured amniocytes may help to obtain a final diagnosis.

Trisomy 12p has an incidence of 1/50,000 births, with most of the cases resulting from balanced parental translocations. The typical dysmorphic face is reminiscent of PKS; a flat face with full cheeks, prominent forehead with high hairline, hypertelorism, short palpebral fissures, short nose with a broad and flat nasal bridge and anteverted nares, long philtrum, thin upper lip vermillion, everted lower lip, prominent chin, and low-set ears. Some cases are associated with tooth defects (peg-shape, anodontia, bifid incisors), short neck, developmental delay, and hypotonia, but no other major defects such as in PKS [39]. There are no skin pigmentation anomalies, and the neurological outcome is usually better. As with PKS, the facial features change with age (coarser, prognathism, malar hypoplasia) [3].

Fryns syndrome is a severe autosomal recessive disorder characterized by diaphragmatic defects, dysmorphic face (similar to PKS), short distal phalanges, pulmonary hypoplasia, and severe developmental delay. Polyhydramnios; clefting; and ocular, urogenital, cerebral, cardiovascular, and gastrointestinal defects may be associated, as well as low hairline and hypertrichosis. A subset of affected individuals have biallelic pathogenic variants in the *PIGN* gene [3].

Sifrim–Hitz–Weiss syndrome (produced by de novo mutations in *CHD4*, a gene located in PKS critical region) is characterized by milder intellectual disability, dysmorphic face (similar to PKS), hypotonia, hearing loss, enlargement of lateral ventricles, and heart defects. Some patients have bone fusions [3].

The goal of this report is to present three new cases of PKS and two cases of trisomy 12p to analyze the differences between entities, to illustrate and discuss some features rarely cited in the literature, and also to present some particularities not cited yet. Moreover, we present alternative methods of diagnosis that could be easily used in daily practice.

## 2. Materials and Methods

The study includes 5 patients: 3 with PKS due to mosaic tetrasomy 12p (as the result of a supernumerary i(12p)) and 2 with trisomy 12p. For each of them, the diagnostic odyssey was different, and the methods used are described in the case presentation.

The informed consent was signed by the parents according to the SPO-MED-52/2013 and PO-MED-17/2013 procedures, approved by the Ethics Commission of Obregia Clinical Hospital of Psychiatry and the Ethics Commission of Saint Mary’s Emergency Children’s Hospital Iasi, Romania (approval No. 691).

## 3. Results

### 3.1. Case 1

The first case is a female aged 3 years (y) and 7 months (mo), the second child of a young, unrelated, apparently healthy couple. There are no similar cases in the family. Pregnancy was uneventful, however fetal sonographic evaluation (2nd trimester) revealed enlarged cerebral ventricles. The child was born by C-section at 39 weeks, with an overgrowth of weight 5310 g (+5.3 SD), length of 57 cm (+4.3 SD), head circumference of 38 cm (+3.3 SD), and Apgar score of 8, with difficult respiratory adaptation immediately after birth. She presented with hypotonia and marked developmental delay (at 3 y 7 mo of age she could raise the head with difficulty, was able to sit with support, but spoke no intelligible words). Myoclonic jerks were noted from the first day of life and intensified after 1 y of age (with EEG expression). More complex seizures were noted after 3 years. The child was suspected of deafness at 2 mo, which was confirmed by ASSR with bilateral moderate sensorineural hearing loss. A hearing aid was inserted at 8 mo and frequent otitis media were recorded in time. No visual (object) reaction was noted at 2 mo, however the ophthalmologic examination was relatively normal (interpreted as cortical blindness). The child was diagnosed at 2 mo with a congenital heart defect. Oxygen desaturation during the night was observed after 1 y of age. Recurrent infections (without fever) in different locations were diagnosed over time (otitis, urinary tract infection, pneumonia). During an episode of pneumonia (at 1 y 3 mo) in the ICU, she was given several albumin infusions, and every time she developed a fever after a few hours. Left hip dysplasia was diagnosed at 1 mo of age. Delayed and abnormal teeth eruption (but all teeth buds were present upon radiography) was noted, as well as slow growth of hair and nails. The mother noted slow bowel movements after 7 mo of age (food diversification started) and days with long periods of very deep sleep after 3 y of age. First clinical genetic examination was performed at 2 mo of age and revealed overgrowth (length +2.8 SD, weight +5.5 SD, head circumference +1.7 SD, followed by growth delay; currently, the body dimensions are in the lower normal range and the head is brachycephalic); fine, depigmented hair; light complexion; marked excess of hair on the forehead and ears (see Figure 1; this later normalized); sparse or fine eyebrows and lashes; hypertelorism; particular iris characteristics (dark blue, except the part around the pupil, which is light blue; pointed pupil with prompt reaction to light); short or broad nose with a depigmented area of skin on the lateral side; long or featureless philtrum; “Pallister lip”; downturned mouth with thin upper lip and everted lower lip; micrognathia (later on normalized); full cheeks; low-set ears; short neck; prominent abdomen with umbilical hernia; hypoplastic external genitalia and anterior displacement of anus; relatively short limbs with acromicria (hands p25, feet p3); soft, puffy skin; small aplasia cutis between vaginal opening and anus; hypotonia; absence of osteotendinous reflexes. Later reevaluations revealed marked developmental delay and no reaction to visual stimuli, sounds, or smell. The child seems to have tactile sense. Temporofrontal baldness and coarse face became evident after 5 mo of age.

Various investigations were performed (hematology: mild microcytic anemia; biochemistry: glucose, liver, and kidney function normal, however lactic dehydrogenase increased repeatedly; ECG: incomplete right bundle branch block; echocardiogram: atrial septal defect, mild pulmonary valve stenosis; abdominal ultrasound: normal; EEG: isolated epileptic spasms when the child is awake, associated with slow biphasic waves, no hypsarrhythmia; cerebral MRI: thin corpus callosum with delayed myelination, anterior fossa not completely formed and disproportionately small (Arnold Chiari type II criteria not accomplished); small pineal gland; eye examination: iris stromal atrophy, mild myopia; ENT examination +BERA, ASSR: moderate hearing loss; endocrine: normal TSH and FT4, low IGF1, high ACTH, normal cortisol; neonatal metabolic screening (plasma): normal; immunology: immunoglobulin A, E, G, M, and serum protein electrophoresis normal).

A standard G-banded karyotype performed on peripheral lymphocytes at 3 mo of age revealed no abnormalities after evaluating 150 metaphases (450 bands).

Microarray-based comparative genomic hybridization (array CGH) analysis of genomic DNA extracted from a buccal swab was conducted using a CGX-HD oligonucleotide array (Perkin Elmer, Turku, Findland). Female genomic DNA by Promega was used as a reference. The CGX-HD array contains 180K oligos but does not cover regions specific to centromeres, heterochromatin, or the short arms of acrocentric chromosomes. The practical average resolution of the array is 200 kb. Data were analyzed using CytoGenomics 2.5 (Agilent, Santa Clara, CA, USA) and Genoglyphix 3.0 (Perkin Elmer, Turku, Findland) software with annotations of Genome Build 37. Array CGH analysis revealed a gain of 33.89 Mb on chromosomal region 12p: arr[GRCh37] 12p13.33-p11.1(190,462−34,078,153) × 3~4 (ISCN 2020), i.e., tetrasomy 12p comprising 218 OMIM-annotated genes (Figure 2). The array CGH profile most likely represents an extra isochromosome for the short arm of chromosome 12.

Having the clinical suspicion of PKS confirmed by array CGH, we extracted DNA from other biological samples (blood, buccal swab, and urine) and performed MLPA using a SALSA MLPA P230 Human Telomere-7 kit (MRC-Holland, Amsterdam, The Netherlands) according to the manufacturer’s instructions. The number of DNA copies was estimated using the Coffalyser.Net software, which calculates the ratio of peak areas in test samples over those of normal controls for each target sequence. We detected the 12p copy number gain (30–50% increased relative peak area of the amplification products of all probes) only in the buccal swab sample (Figure 3). In the clinical context presented above (with major defects associated, highly suggestive for PKS) and due to the different results for different tissues, we interpreted this result as being mosaic tetrasomy 12p, however a FISH test with probes for 12p would be necessary for a definitive diagnosis.

### 3.2. Case 2

The second case is a male aged 16 y, the first child of a young, unrelated, apparently healthy couple. There are no similar cases in the family. Pregnancy evolved with HTA, threatening miscarriage at 14 weeks. The child was born by C-section at 9 mo, with a weight of 3800 g (+1.2 SD), length of 51 cm (+0.5 SD), head circumference of 37 cm (+1.6 SD), Apgar score of 8, and with neonatal jaundice for 3 days. He was diagnosed at birth with hypospadias and bilateral talus valgus. He had frequent unexplained episodes of fever, frequent respiratory infections until age 7 y, temporary teeth extraction at 4 y, and at 8 y of age had adenoidectomy (anesthesia incident noticed). Seizures started after 3.5 mo of age. Marked developmental delay occurred (raised head at 4 y, not able to sit or walk). We first evaluated him at 1.5 mo, when we noticed: normal size; translucid skin with visible collateral circulation on head and abdomen (but without pigmentary anomalies); dry skin on the head and dorsal region; abnormal hair distribution with marked frontotemporal baldness; dysmorphic face (tall, broad, prominent forehead; sparse eyebrows and lashes; small cornea and no reaction to visual stimuli; short nose with anteverted nares; long and smooth philtrum; macrostomia with thin upper lip and everted lower lip; microretrognathia; minor ear defects); short neck; supernumerary nipple; upper limbs with rhizomelic shortening; hands with ulnar deviation of fingers, camptodactyly, and flexed fist; deep palmar and plantar creases; bilateral talus valgus; abnormal genitalia (hypospadias, bifid scrotum, large testes); and marked hypotonia. At 14 mo, we noticed: brachycephaly; thick alveolar ridges; round and mispositioned teeth; deep palmar and plantar creases; shawl scrotum; marked hypotonia; and developmental delay. The last reevaluation was at 16 y, when he had: microbrachycephaly; mild frontotemporal baldness; dysmorphic face (sparse eyebrows and lashes, relatively small cornea, “pointed pupil”, flat zygomatic area, normal nose, short and prominent philtrum, macrostomia, minor ear defects); upper and lower limb spasticity; severe intellectual disability (Figure 4).

Brain CT: Symmetric dilatation of basal cisterns and subarachnoidian space, moderate dilatation of the ventricular system. Babygram (1.5 mo old): Hypoplastic humerus, delayed ossification of the cranial vault.

The first karyotype (from blood, performed at age 1 y) raised the suspicion of an addition on 22q, and in 1 out of 36 metaphases analyzed an i12p was identified (interpreted as an artifact), however no further investigations were possible at that moment. The karyotype from skin fibroblasts (Figure 5) was repeated at 6 y and established the diagnosis of PKS (result: mos47, XY, +i(12p); ish 12pter (telpterx4) (40%)/46, XY (60%)). When we reevaluated the child at age 16 y, all genetic investigations (blood karyotype and MLPA, as well as MLPA using DNA samples extracted from buccal swab, hair root, and urine) were normal.

### 3.3. Case 3

The third patient is a girl aged 6 mo, the first child of healthy, unrelated parents, born at 36 weeks by C-section due to polyhydramnios, with a weight of 2840 g (−1.2 SD), length of 49 cm (0.0 SD), head circumference of 36 cm (+1.5 SD), Apgar score of 4/6/8, however with good postnatal adaptation. Physical examination revealed: normal growth, with a height of 67 cm (+1.0 SD), weight of 6200 g (−1.4 SD), HC of 43 cm (+0.9 SD); dysmorphic face (prominent forehead, frontotemporal baldness, sparse eyebrows, hypertelorism, down slanting palpebral fissures, ptosis, epicanthal folds, broad nasal bridge, long philtrum, thin upper lip, everted lower lip, micrognathia, low-set malformed ears); marked hypotonia; delayed motor development with no head control, no syllables (only sounds), no prehension; and a mental age around 2 mo. The girl also presented epileptic seizures and infantile spasms. Brain MRI was normal and EEG revealed epileptiform discharges.

Patient genomic DNA extracted from peripheral blood was investigated by array CGH using commercial female DNA (Agilent) as a reference and an 8 × 60K human oligonucleotide array platform (Agilent Technologies). Each array grid contains 60K oligos, with an overall median probe spacing of 41 kb. Data analysis was performed with Agilent Cytogenomics software using the reference Genome Build 37. Array CGH analysis revealed a gain of 12p13.33–p11.1, with a size of 34.52 Mb: arr[GRCh37] 12p13.33p11.1 (230,421_34,756,209) × 2~4 (ISCN2020). This gain was interpreted as a mosaic tetrasomy of 12p, thus confirming the PKS diagnosis. The G-banded karyotype performed on peripheral blood revealed no cytogenetic changes. Interphase FISH with 12p telomeric probes confirmed the presence of mosaic 12p tetrasomy in peripheral blood nuclei (Figure 6).

### 3.4. Case 4

The fourth patient is a girl aged 7 mo, the first child of healthy, unrelated parents, born at 32 weeks by C-section, with a weight of 2000 g (+0.3 SD), length of 42 cm (0.0 SD), head circumference of 31.5 cm (+1.6 SD), Apgar score of 7, but with good postnatal adaptation. Physical examination revealed: length of 65 cm (−0.63 SD), weight of 8950 kg (+2.0 SD), HC of 45.5 cm (+2.5 SD), dysmorphic face (prominent forehead, hypertelorism, down-slanting palpebral fissures, partial palpebral ptosis of the left eye, broad nasal bridge, long philtrum, thin upper lip, everted lower lip, micrognathia, low-set malformed ears), muscle hypotonia, delayed motor development (she cannot roll over or sit; she says no syllables, only sounds; she has no prehension, no smile, no interest in toys or objects); her mental age is around 2 mo. The girl also presented gastroesophageal reflux; her abdominal ultrasound was normal.

Array CGH analysis was performed using patient genomic DNA extracted from peripheral blood and Agilent commercial female reference DNA on an 8 × 60K human oligonucleotide array platform (Agilent Technologies), as described above. Genomic profile analysis revealed a deletion of 12p13.33 (0.174 Mb) and an interstitial gain of 12p13.33-p11.22 (28.28 Mb): arr[GRCh37] 12p13.33 (230,421_404,743) × 1, 12p13.33p11.22 (574,727_28,854,069) × 3 (ISCN2020). The genomic profile of this patient revealed a duplication of 12p13.33-p11.22 flanked by a small deletion (Figure 7). While the deletion did not encompass genes considered relevant for the phenotype, the partial trisomy overlaps 12p13.31 region, and among the 233 OMIM genes includes the candidate genes for the PKS phenotype [19]. The G-banded karyotype performed on peripheral blood confirmed the 12p duplication in this patient.

### 3.5. Case 5

The fifth patient is a boy aged 2 y, the first child of healthy, unrelated parents. The pregnancy and birth were uneventful, however his psychomotor development was delayed—he raised his head at 6 mo, sat at 14 mo, walked at 2 y, and said his first syllables at 18 mo. Physical examination revealed: a height of 91 cm (+1.8 SD), weight of 14 kg (+2.0 SD), HC of 49 cm (0 SD), dysmorphic face (prominent forehead, hypertelorism, down slanting palpebral fissures, broad nasal bridge, anteverted nostrils, long philtrum, thin upper lip, prominent and everted lower lip, dysplasia of dental enamel, low-set malformed ears, abnormally folded helix, preauricular pit), 5th finger clinodactyly, wide space between first and second toe, pectus carinatum, and cryptorchidism. Neurological examination showed hypotonia, delayed motor development, speech delay (he says only a few syllables), and severe intellectual disability (a mental age of 8 mo). The psychological evaluation revealed a DQ of 34. The boy also presented recurrent wheezing, gastroesophageal reflux, and atopic dermatitis. Cerebral CT showed mild cortical atrophy. Ophthalmologic examination, heart and abdominal ultrasound, and EEG were normal.

GTG banding showed duplication with inversion for the region of 12p12.3-12p13.3 [46, XY, dup (12) (p13.3p12.3)]. The results were confirmed by FISH with painting and subtelomeric probes for chromosome 12 (Aquarius, Cytocell).

## 4. Discussion

All patient data are described in detail for every patient, and both the literature data and patient information are summarized in Table 1 to facilitate discussions.

We identified some novel and rarely cited clinical features concerning facial hair distribution, eye structure, skin and connective tissue, immune defense, autonomic and sensory dysfunction, and seizures—particularities that will be discussed in detail below.

The striking facial aspect noticed at the first clinical genetic evaluation of case 1 (2 mo of age) consisted of a marked excess of hair on the forehead and ears (Figure 1), features that disappeared by the second evaluation at 5 mo. This could represent a particular variant of lanugo, as it was not generalized (as with classical cases) and disappeared in the first months of life. However, classical lanugo is found in premature babies and could be a sign of severe malnutrition, which is not the case in our patient. Moreover, hirsute face and distal digital hypoplasia (features present in case 1) are features rarely associated in Fryns syndrome and not in PKS [3]. This observation raises the question of whether PKS and PKS-like disorders in fact represent a spectrum ranging from Fryns syndrome (severe end) to PKS (moderate), trisomy 12p (moderate–mild), and Sifrim-Hitz-Weiss syndrome (mild end). Further studies are needed.

The eye malformation identified in case 1 consisted of the particular color of the iris (dark blue, except the part around the pupil, which is light blue) and miosis (“pointed pupil”), however with prompt reaction to light (Figure 1). This aspect has been constant over time and is due to the stromal atrophy of the iris, as described by the patient’s ophthalmologist. The “pointed pupil” symptom was also noticed in case 2 but not in case 3. Iris atrophy has rarely been described [6,40], however it probably should be investigated in children with suspicion of PKS.

Other craniofacial changes with age were provided by case 2, who had 16 years of follow-up: the skull (normal initially) became markedly brachycephalic in time, frontotemporal baldness (very marked in the beginning) diminished with age but was still present for an attentive eye, zygomatic areas became very flat in time, and the nose changed totally (from flat and broad with anteverted nostrils to a normal, pointed nose). The perioral region and ears remained unchanged. Occipital and zygomatic flattening was also noticed in case 1 (see Figure 1). A possible explanation for occipital flattening over time could be prolonged periods spent lying in bed in the same position due to severe hypotonia. Studies in the literature mention that with age, the face becomes coarser and the chin becomes more prominent. In our cases, face coarseness did not change over time (present in case 1, absent in case 2), however prognathism was noticed in case 2 at an older age. Cases 3–5 have no follow-up. Malar flattening is cited in the literature [3], however very few authors mention it. Because we have noticed this in two out of three PKS cases, we suggest the use of dedicated software for photo analysis could provide the entire description of phenotype changes over time.

Patient 1’s skin is soft and puffy, which is the reason why the child was suspected and investigated for hypothyroidism. Thyroid investigations have been repeatedly normal. However, such soft, doughy skin (which later became translucent, with visible blood vessels; see Figure 1) is associated with iris stromal atrophy, umbilical hernia, hip dislocation, and delayed tooth eruption (case 1). Case 2 presents translucid skin with visible collateral circulation, which made us think of a possible connective tissue defect, however further studies are necessary to confirm this hypothesis. Moreover, ectodermal derivatives are also abnormal (lanugo described above; temporofrontal baldness; fine or sparse eyebrows and lashes; slow-growing, fine, depigmented hair; slow-growing nails; abnormal tooth eruption), with some of these features having not yet been reported in the literature.

Immune defects are not documented in the literature on PKS so far. Patient’s 1 and 2 have experienced repeated episodes of infection. However, these infections are difficult to identify, as child 1 developed no fever during infection and child 2 had frequent unexplained episodes of fever, suggesting hypothalamic dysfunction. As our initial immunological tests were normal, lymphocyte subpopulation analysis should be performed. The autonomic dysfunction is also supported by the presence of days with long periods of very deep sleep, which in case 1 (“like being in a coma”) was noticed by the mother. Moreover, the immune phenotype seems to be associated with a particular immune reaction, such as with case 1 developing fever after being given albumin infusion. Such particularities have not yet been described in PKS and probably should be investigated in confirmed cases. The mild microcytic anemia of the child could be due to a nutritionally deficient diet, but also to repetitive infections.

Patients 1–3 present the typical growth pattern cited in the literature for some of the PKS cases [16,17], including initial macrosomia, followed by postnatal growth deficit evident by one year of age. The explanation provided in the literature [41] is that PKS individuals have elevated IGFBP2 that binds IGF1. The reduced IGF1 available will lead to growth deficit. We were not able to evaluate IGFBP2 in case 1, but because we found low IGF1, we expect that our patient has elevated IGFBP2, explaining the noticed postnatal growth deficit. Moreover, we found a small pineal gland in brain MRI and high ACTH. We appreciate that more in-depth endocrine investigations should be performed to elucidate the entire mechanism and provide eventual therapeutic options. An interesting aspect is that IGFBP2 has also been involved in diabetes and cancer. Periods of lethargy noticed by the mother in case 1 could be related to the dysfunctional brain, however also to episodes of hypoglicemia, a reason why we plan to monitor blood sugar levels in our patient.

Classical sensory impairments cited as being sometimes associated with PKS are deafness and (central) blindness [25,36]. Both are present and relatively severe in patient 1, contributing to the severe developmental delay of the child. Moreover, we have tried to test her ability to smell and found no reaction, however we think this is a very difficult test for the age of 1 y and 5 mo and we should test her again when she has grown older. Cases 2–3 do not seem to have sensory impairments but were not tested in detail.

Myoclonic jerks have been noted in case 1 from the first day of life, which intensified after 1 y of age (with EEG expression). Moreover, seizures have been a constant feature in all of our PKS cases. Published data mention that seizures are extremely rare neonatally and they become evident later in life [11]. We suppose that seizures could be very mild in the beginning, a reason why they are noted only later. Detailed anamnesis referring to mild types of seizures (e.g., myoclonic jerks) should be performed in any confirmed PKS case.

Contractures have been present from an early age in case 2, which progressed to marked spasticity of upper and lower limbs. These features have rarely been described in the literature.

Regarding diagnostic laboratory methods, the initial karyotype (from cultured peripheral blood) was normal for case 1, however due to the suggestive PKS features, we performed array CGH using DNA extracted from a buccal swab. This test confirmed the presence of 12p tetrasomy (Figure 2). However, since this diagnostic test is relatively expensive for developing countries, we performed telomeric MLPA on DNA samples from different sources (blood, buccal swab, urine) to prove the existence of 12p tetrasomy using an alternative, cheaper method. Surprisingly, 12p tetrasomy was identified only in the buccal swab, while for blood and urine samples the test was normal. We offer two possible explanations for this result: either the 12p tetrasomy is only present in some organs, or because we performed this test when the child was already 1 y and 5 mo old of age, by this time abnormal cells could have disappeared due to the specific selection against i(12p) positive cells. Moreover, MLPA investigation is a very accessible test. We suggest MLPA (follow-up kit P230-B1) should be done in children under suspicion of PKS using samples from different tissues (buccal swab should definitely be one of them). In case 2, the first karyotype identified the isochromosome, but because it was found only in 1 out of 36 cells read, it was considered an artifact. Thinking retrospectively, using interphasic FISH with 12p probes (after considering the clinical features and the karyotype) would probably have provided the diagnosis earlier. Moreover, a comprehensive clinical evaluation is essential for infants with multiple congenital anomalies and severe developmental delay. For such cases, blood analysis may in fact be enough to diagnose PKS within the first months of life, especially in neonates.

In our experience, the association of dysmorphic face (prominent forehead, frontotemporal baldness, long philtrum, everted lower lip) with marked developmental delay and seizures is highly suggestive for the diagnosis of PKS, and MLPA (follow-up kit P230-B1) should be performed as a first intention test using DNA samples from blood and a buccal swab. If the clinical diagnosis is not as straightforward, an array CGH using a DNA sample extracted form a buccal swab would be the best method to provide the diagnosis. In confirmed cases, sensorial dysfunction (blindness and hearing loss) should be investigated. Infections seem to be common, meaning the people taking care of these patients should be aware of this finding.

Trisomy 12p is defined by similar facial characteristics to PKS (but without frontotemporal baldness), associated with hypotonia and developmental delay, however no other major anomalies are associated. The cases presented here are typical and the diagnoses have been clearly established using array CGH. However, because array CGH is a laborious and relatively expensive method, we recommend using subtelomeric MLPA first (mainly based on the consideration that many cases of trisomy 12p could be the result of a balanced abnormality in one of the parents, meaning that the 12p duplication could be associated with a deletion in a different chromosome). For confirmed cases, as the trisomy could be inherited, a karyotype is indicated for both parents.

Differential diagnosis between PKS, Fryns, and Sifrim-Hitz-Weiss syndrome (see Table 1) is an important issue, as the syndromes are clinically very similar, however the genetic causes are totally different. We suggest that these aspects should be evaluated before proceeding to genetic testing.

## 5. Conclusions

We have identified features that have not yet been or that are rarely reported in PKS studies i.e., marked excess of hair on the forehead and ears in the first months of life, a particular eye disorder (abnormal iris color with pointed pupil), connective tissue defects, repeated episodes of infection and autonomic dysfunction, endocrine malfunction as a possible cause of postnatal growth deficit, more complex sensory impairments, and mild early myoclonic jerks.

MLPA (follow-up kit P230-B1) and array CGH using DNA extracted from a buccal swab are reliable methods of diagnosis in PKS and we recommend them as the first intention diagnostic tests.

## Figures and Tables

**Figure 1 genes-12-00811-f001:**
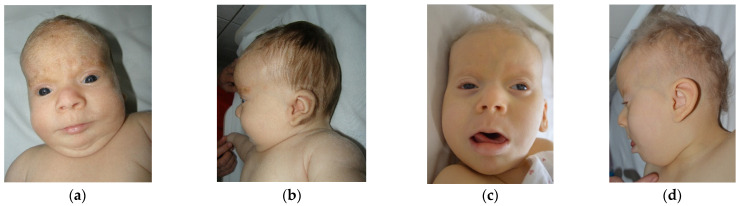
Facial characteristics of patient 1: (**a**,**b**) 2 mo of age; (**c**,**d**) 3 y and 7 mo of age.

**Figure 2 genes-12-00811-f002:**
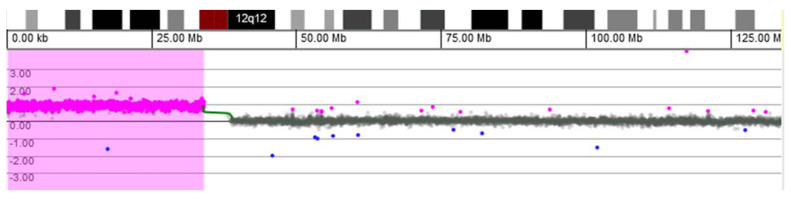
Array CGH result for patient 1—DNA extracted from a buccal swab.

**Figure 3 genes-12-00811-f003:**
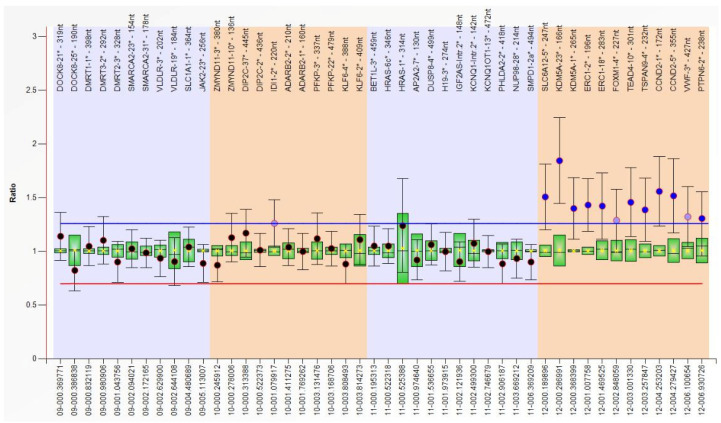
MLPA result for patient 1—DNA extracted from buccal swab. * new version probe P230.

**Figure 4 genes-12-00811-f004:**
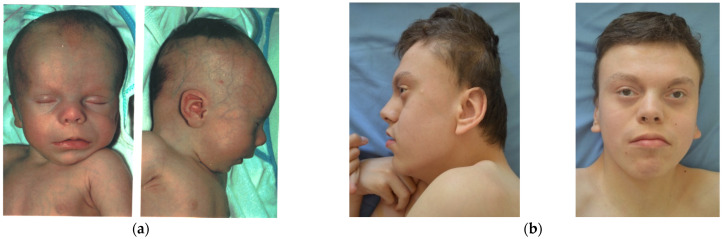
Clinical features of patient 2: (**a**) 1.5 mo old; (**b**) 16 y old.

**Figure 5 genes-12-00811-f005:**
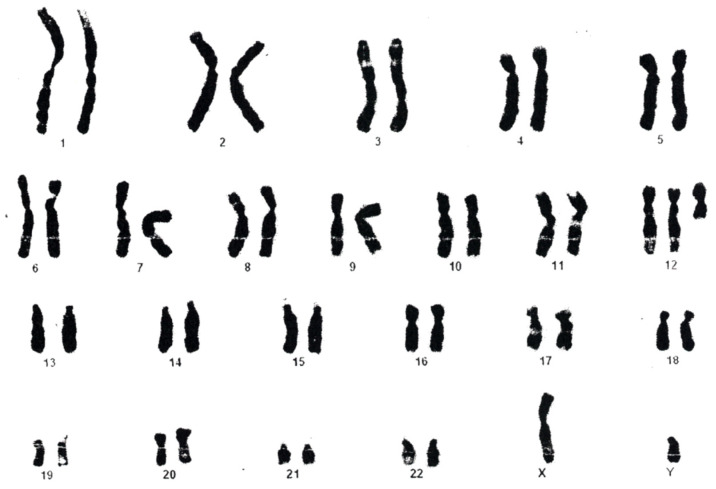
Karyotype for patient 2.

**Figure 6 genes-12-00811-f006:**
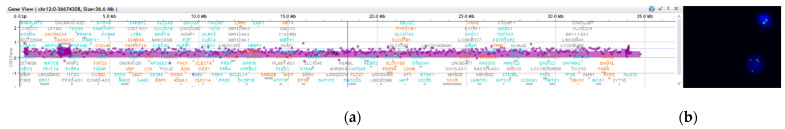
Genetic tests for patient 3: (**a**) array CGH result—DNA extracted from a peripheral blood; (**b**) interphase FISH—peripheral blood.

**Figure 7 genes-12-00811-f007:**
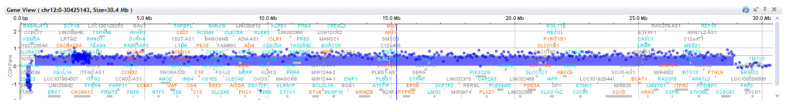
Genetic tests for patient 4. Array CGH result—DNA extracted from a peripheral blood.

**Table 1 genes-12-00811-t001:** Literature data (columns 2–4 for differential diagnosis) and patient data (columns 5–9).

Clinical Features	Pallister Killian sdr.	Trisomy 12p	Fryns sdr.	Sifrim–Hitz–Weiss sdr.	PKS 1	PKS 2	PKS 3	Tri 12p 1	Tri 12p 2
Frontotemporal baldness	+	−	−	−	+	+	+	−	−
Coarse face	++	−	+	+	+	−	+	−	−
Prominent forehead	+	+	−	+	+	+	+	+	+
Flat occiput	+	−	−	−	−→+	+	−	−	−
Long philtrum	+	+	−	+	+	+→−	+	+	+
Macroglossia	+	+	−	−	+	−	+	−	−
Cleft lip/palate	+/−	+	+	+	−	−	−	−	−
Everted lower lip	+	+	−	−	+	+→−	+	+	+
Accessory nipples	+	+	−	−	−	+	−	−	−
Focal aplasia cutis	+	−	−	−	+	−	−	−	−
Hypo/hyperpigmented areas	+	−	−	−	+	−	−	−	−
Congenital diaphragmatic hernia	+	−	+	+	−	−	−	−	−
Deafness	+	+	−	+	+	−	−	−	−

Only features that make a difference between syndromes are presented; + present; − not present.

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
