# Peer review of "Pallister–Killian Syndrome versus Trisomy 12p—A Clinical Study of 5 New Cases and a Literature Review"

_genes, 2021, doi:10.3390/genes12060811_

Round 1
Reviewer 1 Report
The paper by Aurora Arghir et al. presents a series of 3pts w/tetrasomy 12p and 2pts w/trisomy 12p. The potential advantage of the publication is: 1. detailed clinical characteristics of all the cases, 2. comparison of tetrasomy vs trisomy 12p.
Major comments:
- blood analysis may in fact be enough within the first months of life to diagnose PKS, especially in neonates; that should be pointed out in the paper
- the authors use the terms buccal swab AND saliva; saliva is generally thought of as representing blood lymphocytes, whereas the swab is a different tissue comparable to skin fibroblasts; did some children actually spit the saliva into the tube or was the buccal swab taken in all cases? with the latter being true the term saliva should not be used...
- the authors show features they claim might be novelties, however: iris transillumination with pupil reacting to light was already reported by Ticho, 2010; myoclonic seizures were reported by Giordano et al., 2012 (both refs should be added within the text); I agree though that myoclonic jerks could have been present since day 1 and that is at least SOME novelty; please explain what is meant by connective tissue dysfunction (given it is very common in children)
- DV agenesis might be a marker of PKS, yet already in more than a quarter of prenatal cases it is associated with cardiac and extracardiac anomalies [Pacheco et al.], thus PKS may be within this 1/4 as well
- CGX-HD platform has 40kb resolution; which platform was used: CGX or CGX-HD?
- array technology is in fact so widely used (even with financial restrictions) that it has literally outclassed MLPA; thus a claim that MLPA is still 'alive' in diagnostics is unsupported should only be used whenever there is a strong suspicion of a certain syndrome, which is not the case in most instances of chromosomal aberrations; that should be corrected for within the text
- it could be of use if the authors were to compare how much of the mosaic (percentage-wise) was detected through array in all cases
- please explain 365-367: how does fever relate to dev delay?
- what causes Fryns syndrome?; explain in the text
Minor comments:
- add centiles or SD
- correct: macrostomia, ECG, pointed pupil
Author Response
Reviewer 1
Dear reviewer,
Thank you for all your comments. They were really useful and definitely contributed to the improvement of the manuscript. We shall take them in order:
Major comments:
- blood analysis may in fact be enough within the first months of life to diagnose PKS, especially in neonates - we have pointed this out in the paper.
- the authors use the terms buccal swab AND saliva; saliva is generally thought of as representing blood lymphocytes, whereas the swab is a different tissue comparable to skin fibroblasts; did some children actually spit the saliva into the tube or was the buccal swab taken in all cases? with the latter being true the term saliva should not be used... – we have replaced saliva with buccal swab.
- the authors show features they claim might be novelties, however: iris transillumination with pupil reacting to light was already reported by Ticho, 2010; myoclonic seizures were reported by Giordano et al., 2012 (both refs should be added within the text); I agree though that myoclonic jerks could have been present since day 1 and that is at least SOME novelty; please explain what is meant by connective tissue dysfunction (given it is very common in children) –Thank you for your kind observations. We have cited the articles mentioned and we replaced the term dysfunction with defect that seems more appropriate for connective tissue. The clinical description should be a key part of the manuscript, so the appropriate use of the terms is really important.
- DV agenesis might be a marker of PKS, yet already in more than a quarter of prenatal cases it is associated with cardiac and extracardiac anomalies [Pacheco et al.], thus PKS may be within this 1/4 as well we have cited the article mentioned, and included more detailes in the rows 106-109;
- CGX-HD platform has 40kb resolution; which platform was used: CGX or CGX-HD? It is CGX-HD. The test was done in 2013 and at that time the platform was having 200kb resolution;
- Array technology is in fact so widely used (even with financial restrictions) that it has literally outclassed MLPA; thus a claim that MLPA is still 'alive' in diagnostics is unsupported should only be used whenever there is a strong suspicion of a certain syndrome, which is not the case in most instances of chromosomal aberrations; that should be corrected for within the text – we have included changes according to these valuable indications. Thank you!
- It could be of use if the authors were to compare how much of the mosaic (percentage-wise) was detected through array in all cases – Unfortunately we couldn’t do this because the reports do not include data;
- please explain 365-367: how does fever relate to dev delay? – we have excluded these misleading sentences;
- what causes Fryns syndrome?; explain in the text –we have included detailed information on the differential diagnosis in rows 121-130. Thank you for asking this!
Minor comments:
- add centiles or SD – done; SD included;
- correct: macrostomia, ECG, pointed pupil – done;
Hope we have touched all the points you asked us to change.
If there are any other changes you consider we should make, please let us know.
Yours sincerely,
All the authors
Reviewer 2 Report
The manuscript might be a useful contribution especially in delineating the trisomy 12p syndrome profile. However, as the title states “literature review” I would suggest to perform a more accurate analysis of literature and edit accordingly the manuscript (eg. high IGFBP2 levels have been related to post-natal growth deficiency in PKS patients while you say in discussion that the typical growth pattern of PKS patients is reported in literature, but “laboratory investigations are not mentioned”; recurrent infections in PKS patients have already been reported; in table 1 cleft palate has been reported in PKS patients while you stated it is not present in PKS patients etc…).
In terms of PKS clinical history the manuscript focus on some clinical issues that still need to be characterized.PKS patients are known to have a clinical history of recurrent infections, especially respiratory recurrent infections, but to date an immunological basis for these infections has not be demonstrated (it is likely due, at least in part, to hypotonia). It would be nice if you report on serum immunological examination performed on their patients (es. Lymphocyte subpopulation analysis, quantitative immunoglobulin tests etc).
Also, many families describe sleep problems, characterized by inability to distinguish day from night, and sleep apneas and it is already known PKS individuals tend to sleep more hours per day, range from 9.5 to 15 hrs per day. This is particularly true for patient with visual and hearing impairment as for case 1 in the paper.
In terms of PKS cytogenetics diagnosis the manuscript does not add any real new insight to what already known, but comparing results on blood and buccal swab sample, may be useful to validate a less invasive second tier for PKS diagnosis. Conlin et al., have already shown that array-based cytogenetic methodologies may be more sensitive than standard G-banding karyotyping methods in identifying tetrasomic cells also in peripheral blood especially under 12 months. After the first year of life is definitely better search for tetrasomic cells on different tissue as swab by array based cytogenetic methodology. Below some useful references on PKS prenatal and post-natal profile, and its differential diagnosis (including the trisomy 12p) that you may use to update the manuscript and to build up a comparison with your patients as well as to arrange a differential diagnosis with trisomy 12p.
Izumi, K., Kellogg, E., Fujiki, K., Kaur, M., Tilton, R. K., Noon, S., ... & Krantz, I. D. (2015). Elevation of insulin‐like growth factor binding protein‐2 level in Pallister–Killian syndrome: Implications for the postnatal growth retardation phenotype. American Journal of Medical Genetics Part A, 167(6), 1268-1274.
Salzano, E., Raible, S. E., & Krantz, I. D. (2021). PALLISTER–KILLIAN SYNDROME. Cassidy and Allanson's Management of Genetic Syndromes, 717-733.
Salzano, E., Raible, S. E., Kaur, M., Wilkens, A., Sperti, G., Tilton, R. K., ... & Krantz, I. D. (2018). Prenatal profile of Pallister‐Killian syndrome: Retrospective analysis of 114 pregnancies, literature review and approach to prenatal diagnosis. American Journal of Medical Genetics Part A, 176(12), 2575-2586.
Author Response
Reviewer 2
Dear reviewer,
Thank you for appreciating the manuscript and for suggestions that improved the quality of the manuscript, also for the prompt and concise evaluation. Please find your suggestions and our answers below:
The manuscript might be a useful contribution especially in delineating the trisomy 12p syndrome profile. However, as the title states “literature review” I would suggest to perform a more accurate analysis of literature and edit accordingly the manuscript (eg. high IGFBP2 levels have been related to post-natal growth deficiency in PKS patients while you say in discussion that the typical growth pattern of PKS patients is reported in literature, but “laboratory investigations are not mentioned”; recurrent infections in PKS patients have already been reported; in table 1 cleft palate has been reported in PKS patients while you stated it is not present in PKS patients etc…). We have updated the literature review according to your observations and added a new paragraph in rows 411-420. The new references have been included, thank you for suggesting them!
In terms of PKS clinical history the manuscript focus on some clinical issues that still need to be characterized.PKS patients are known to have a clinical history of recurrent infections, especially respiratory recurrent infections, but to date an immunological basis for these infections has not be demonstrated (it is likely due, at least in part, to hypotonia). It would be nice if you report on serum immunological examination performed on their patients (es. Lymphocyte subpopulation analysis, quantitative immunoglobulin tests etc). All the investigations were normal, reason why they were not included; we have added the normal results in rows 196-197;
Also, many families describe sleep problems, characterized by inability to distinguish day from night, and sleep apneas and it is already known PKS individuals tend to sleep more hours per day, range from 9.5 to 15 hrs per day. This is particularly true for patient with visual and hearing impairment as for case 1 in the paper. Thank you for observing this. It is indeed happening so with the severely affected individuals. We have added a paragraph in rows 418-420.
In terms of PKS cytogenetics diagnosis the manuscript does not add any real new insight to what already known, but comparing results on blood and buccal swab sample, may be useful to validate a less invasive second tier for PKS diagnosis. Conlin et al., have already shown that array-based cytogenetic methodologies may be more sensitive than standard G-banding karyotyping methods in identifying tetrasomic cells also in peripheral blood especially under 12 months. After the first year of life is definitely better search for tetrasomic cells on different tissue as swab by array based cytogenetic methodology. Below some useful references on PKS prenatal and post-natal profile, and its differential diagnosis (including the trisomy 12p) that you may use to update the manuscript and to build up a comparison with your patients as well as to arrange a differential diagnosis with trisomy 12p.
Izumi, K., Kellogg, E., Fujiki, K., Kaur, M., Tilton, R. K., Noon, S., ... & Krantz, I. D. (2015). Elevation of insulin‐like growth factor binding protein‐2 level in Pallister–Killian syndrome: Implications for the postnatal growth retardation phenotype. American Journal of Medical Genetics Part A, 167(6), 1268-1274.
Salzano, E., Raible, S. E., & Krantz, I. D. (2021). PALLISTER–KILLIAN SYNDROME. Cassidy and Allanson's Management of Genetic Syndromes, 717-733.
Salzano, E., Raible, S. E., Kaur, M., Wilkens, A., Sperti, G., Tilton, R. K., ... & Krantz, I. D. (2018). Prenatal profile of Pallister‐Killian syndrome: Retrospective analysis of 114 pregnancies, literature review and approach to prenatal diagnosis. American Journal of Medical Genetics Part A, 176(12), 2575-2586.
Thank you again for your valuable suggestions! We have added these references and reformulated the paragraphs between rows 458-467.
Hope we have touched all the points you asked us to change.
If there are any other changes you consider we should make, please let us know.
Yours sincerely,
All the authors
Round 2
Reviewer 1 Report
Minor revision:
102: explain abbreviations BP nad HC
125: should be: 'Subset of affected individuals'
150: should be: '+4.3 SD'
Should be 'Sifrim-Hitz-Weiss'
Author Response
Reviewer 1
Dear reviewer,
Thank you for suggestions that improved the quality of the manuscript, also for the prompt and concise evaluation. Please find your suggestions and our answers below:
Minor revision:
102: explain abbreviations BP nad HC we have included the descriptions of the abbreviations as suggested.
125: should be: 'Subset of affected individuals'. Thank you for a better formulation than ours. We have included it.
150: should be: '+4.3 SD' We have added “+”. Thank you for noticing it.
Should be 'Sifrim-Hitz-Weiss' We have excluded “l”. Sorry for misspelling
Hope we have touched all the points you asked us to change.
Yours sincerely,
All the authors
Reviewer 2 Report
Overall, the manuscript has been reviewed according to my suggestions.
Some minor edits together with some English polishing still need to be made.
Minor comments:
Line 109: … “should alert for PKS and indicate amniocentesis” … As you said, ductus venosus agenesis is an aspecyfic prenatal marker for several genetic conditions potentially investigable by villo and/or amniocentesis. Thus, I would delete "indicate amniocentesis", just leaving in the sentence “should alert for PKS”.
Line 110: “Amniocentesis and further analysis of amniocytes provide the final diagnosis”.
Unfortunately, it is not so easy to detect the iso12p on amniocytes. I would suggest to replace this sentence with something like that “Array CGH on genomic DNA extracted from uncultured amniocytes may help to get the final diagnosis”.
Line 150, 232: For birth parameters (and in general under 2 years of age) I would use length rather than height
Line 375: “A possible explanation for occipital flattening in time could be prolonged lying in bed in the same position due to severe intellectual disability” I would change as follow “A possible explanation for occipital flattening in time could be prolonged lying in bed in the same position due to severe hypotonia”
Line 397: “The immune particularities identified in our cases consist of an immune deficit (present in both cases 1 and 2), the children having repeated episodes of infection” I would say just that patients 1 and 2 experienced repeated episodes of infection, as their blood/serological examinations do not support an immune deficit (I guess lymphocyte subpopulation analysis have not been performed). And I would modify accordingly also the conclusion paragraph.
Also, as you cite Silfrim-Hitz-Weiss Syndrome among conditions that need to be differentiate from PKS in the introduction, you should add it in table 1 and consider it with Fryns S. and trisomy 12p S. in the discussion as well.
Author Response
Reviewer 2
Overall, the manuscript has been reviewed according to my suggestions.
Some minor edits together with some English polishing still need to be made.
Thank you for all your comments. They were really useful and definitely contributed to the improvement of the manuscript. We shall take them in order:
Minor comments:
Line 109: … “should alert for PKS and indicate amniocentesis” … As you said, ductus venosus agenesis is an aspecyfic prenatal marker for several genetic conditions potentially investigable by villo and/or amniocentesis. Thus, I would delete "indicate amniocentesis", just leaving in the sentence “should alert for PKS”. We have included changes according to these valuable indications. Thank you!
Line 110: “Amniocentesis and further analysis of amniocytes provide the final diagnosis”.
Unfortunately, it is not so easy to detect the iso12p on amniocytes. I would suggest to replace this sentence with something like that “Array CGH on genomic DNA extracted from uncultured amniocytes may help to get the final diagnosis”. We appreciate this sentence as being much more comprehensive than the one included in the manuscript. Thank you for suggesting it!
Line 150, 232: For birth parameters (and in general under 2 years of age) I would use length rather than height. Thank you for your pertinent observation! We have replaced the formulation for all the situations under 2 years of age.
Line 375: “A possible explanation for occipital flattening in time could be prolonged lying in bed in the same position due to severe intellectual disability” I would change as follow “A possible explanation for occipital flattening in time could be prolonged lying in bed in the same position due to severe hypotonia” We have changed the sentence with your suggestion that seems more accurate. Thank you!
Line 397: “The immune particularities identified in our cases consist of an immune deficit (present in both cases 1 and 2), the children having repeated episodes of infection” I would say just that patients 1 and 2 experienced repeated episodes of infection, as their blood/serological examinations do not support an immune deficit (I guess lymphocyte subpopulation analysis have not been performed). And I would modify accordingly also the conclusion paragraph. We have reformulated the paragraphs referring to immune particularities (lines 30, 394-397, 478-479) according to your valuable suggestions.
Also, as you cite Silfrim-Hitz-Weiss Syndrome among conditions that need to be differentiate from PKS in the introduction, you should add it in table 1 and consider it with Fryns S. and trisomy 12p S. in the discussion as well. We have added information regarding Sifrim-Hitz-Weiss syndrome both in table 1 and discussions. Thank you for suggesting it!
Hope we have touched all the points you asked us to change.
Yours sincerely,
All the authors